# Psychophysiological and psychological responses of touching plant behavior by tactile stimulation according to the foliage type

**Seo-Hyun Kim, Sin-Ae Park** [ORCID]*

Department of Bio and Healing Convergence, Graduate School, Konkuk University, Seoul, Republic of Korea

* sapark42@konkuk.ac.kr

## Abstract

Urbanization-related stress has spurred interest in natural therapies, such as horticultural therapy, which leverages multisensory exposure to plants to enhance well-being through physical, psychological, and cognitive benefits. This study aimed to measure and compare the psychophysiological and psychological responses to tactile stimuli through plant contact based on the foliage type. Thirty adults (average age: $24.86 \pm 2.68$) participated in the study, and the foliage was categorized into four groups: soft (e.g., *Stachys byzantina, Adiantum raddianum,* and *Asparagus plumosus var. nanus*), smooth (e.g., *Peperomia obtusifolia, Ficus benghalensis,* and *Epipremnum aureum*), stiff (e.g., *Chamaeshparis thyoides Red Star, Platycladus orientalis,* and *Cupressus macrocarpa*), and rough (e.g., *Rhapis excelsa, Nephrolepis cordifolia* 'Duffii', and *Ardisia pusilla* 'Variegata') plant groups. The participants touched the plants for 90 s, and the concentration of oxyhemoglobin (oxy-Hb) in the prefrontal cortex (PFC) was measured using functional near-infrared spectroscopy (fNIRS). Additionally, a semantic differential method (SDM) evaluation tool was used to assess the psychological responses of each treatment group. When comparing the four tactile treatment groups (soft, smooth, stiff, and rough), the oxy-Hb concentration in the PFC area was lowest during tactile stimulation of smooth plants and highest during soft plant stimulation. Sex-based comparison of oxy-Hb concentrations showed significant differences in the overall PFC area for all four tactile treatment groups in males ($p < 0.001$). Specifically, when touching soft plants, the oxy-Hb concentration in females was significantly lower than that in males ($p < 0.001$). According to the SDM, the tactile stimulation of soft and smooth plants elicited the most relaxation, comfort, and favorable responses ($p < 0.001$). When touching smooth plants, the oxy-Hb concentration of the participant was the lowest, and according to the SDM, they reported the most soothing response. Summarily, the participants in the smooth plant group exhibited a trend of decreased oxy-Hb concentrations and concurrently experienced a sense of psychological stability. We established those tactile stimuli based on foliage texture resulted in different psychophysiological and psychological responses depending on the plant treatment group and sex.

**Data availability statement:** All relevant data are within the manuscript and its Supporting Information files.

**Funding:** This work was carried out with the support of the "Cooperative Research Program for Agriculture Science and Technology Development (Project No.:RS-2021-RD009877)", Rural Development Administration, Republic of Korea.

**Competing interests:** The authors have declared that no competing interests exist.

## Introduction

Rapid urbanization has led to an increase in human stress levels, as high-density environments and artificial surroundings challenge individuals' mental well-being [1]. Urban living has been shown to affect brain function, particularly within the ventromedial prefrontal cortex (vmPFC), a key area involved in regulating emotional responses, where urban-induced stress may amplify negative affective responses and heighten stress-related neural activity [2,3]. To mitigate such urban stressors, interest in nature-based therapies, such as horticultural therapy, has risen, as these approaches incorporate natural elements that offer both psychological and physiological benefits [4].

Horticultural therapy, a form of 'Green Care,' involves tailored gardening activities guided by experts to match the characteristics of the participants. It aims to enhance mental and physical well-being through structured plant-related activities [5,6]. Positive effects across various dimensions physical, psychological, cognitive, social, behavioral, and educational have been documented, with participants in therapeutic agriculture commonly experiencing physical benefits, such as improved health and agricultural skills, as well as psychological gains, including increased self-esteem, emotional stability, quality of life, and trust in others [7–9]. Furthermore, gardening activities allow individuals to engage with natural elements even within urban settings, facilitating interactions with nature despite environmental constraints [10].

Research on sensory experiences in horticultural therapy has traditionally focused on visual and olfactory stimuli, which have been shown to have beneficial effects on mental health and cognitive functioning. For instance, exposure to plants has been linked to decreased oxyhemoglobin (oxy-Hb) concentrations in the prefrontal cortex (PFC), a physiological marker of stress reduction and mental relaxation [11,12]. Visual exposure to green environments has been associated with increased alpha and beta brain waves [13], along with reduced blood pressure, supporting the role of plants in fostering relaxation [14]. However, although studies on visual and olfactory stimuli provide strong support for the benefits of plant-based interactions [15,16], tactile engagement with plants, particularly through touch, remains relatively unexplored in terms of its psychophysiological impacts.

Tactile stimulation is a powerful sensory modality that can evoke profound emotional responses and influence well-being. The skin, as the body's primary sensory interface, perceives a wide range of textures and pressures, facilitating the release of neurochemicals like oxytocin and serotonin, which play critical roles in social bonding and mood regulation [17,18]. Prior studies have shown that tactile stimulation with soft or smooth textures is generally associated with positive affective responses and can reduce stress markers such as cortisol [19]. For example, tactile experiences involving soft, natural materials have been found to elicit greater feelings of comfort and relaxation compared to rough or synthetic materials, as measured through both self-report assessments and physiological markers such as heart rate variability (HRV) [20,21]. Further research has demonstrated that touching plants with smooth, soft textures can lower oxy-Hb levels in the PFC, suggesting an inherent calming effect like that seen with visual plant exposure [22].

Studies in the field of psychophysiology suggest that the tactile experience of different textures can evoke variable neural responses. Soft, smooth textures, for instance, activate regions in the prefrontal cortex associated with positive affect and social engagement, while rough textures can trigger the amygdala, potentially activating stress or discomfort responses [23,24]. Additionally, recent findings indicate that individuals' prior experiences and cultural associations with nature may influence their physiological and psychological responses to plant-based tactile stimuli. This underlines the potential for tactile stimulation to not only alleviate stress but also enhance psychological stability through a perceived connection with nature [25,26].

Despite these insights, research on the psychophysiological effects of tactile stimulation specifically in horticultural settings is limited. Given the unique textures and sensory properties of different plant types, investigating how tactile interaction with foliage affects the brain and emotional responses is essential for expanding our understanding of nature-based therapies. Therefore, this study examines the psychophysiological responses to tactile stimulation with four types of plant textures (soft, smooth, stiff, and rough) using functional near-infrared spectroscopy (fNIRS) to measure prefrontal cortex activity and the Semantic Differential Method (SDM) for self-reported psychological responses. By examining how different foliage textures influence psychophysiological responses, this research aims to enhance our understanding of the therapeutic potential of tactile plant interactions and provide empirical evidence to support the development of sensory-focused horticultural therapy programs.

## Materials and methods

### Participants

This study recruited a cohort of 30 adults in their 20s (15 men and 15 women) based on the assumption that the population from which the sample was drawn followed a normal distribution, as suggested by previous psychophysiological research with a sample size of 30 or more participants engaged in horticultural activities [27]. To recruit participants, notices containing information about the study were posted in schools and libraries near Gwangjin-gu, Seoul, Korea, starting from June 19, 2023, until the end of the recruitment period on July 7, 2023. The selection criteria included individuals with the inclusion criteria comprised individuals exhibiting right-hand dominance [28] and lacking plant allergies. Exclusions were made for individuals currently afflicted by specific illnesses [29] and those experiencing impairments in hand function. Additionally, the participants were requested to fast for 3 h before the experiment to minimize the influence of caffeine [30]. Participants who expressed interest in the study were explained the research purpose and procedures. They were selected based on their voluntary expression of interest and the on-site signing of a consent form. This study was approved by the Institutional Review Board of Konkuk University (Approval No. 7001355-202303-HR-649).

### Experimental environment

The experiment was conducted in an experimental space on the campus of Konkuk University. Following the International Facility Management Association standards for workspace area, the internal space of the experimental space was set to 2.0 m × 2.0 m. The indoor environment was controlled according to the recommendations of the American Society of Heating, Refrigerating, and Air-Conditioning Engineers, with a temperature range of 23–26°C, relative humidity of $30 \pm 10\%$, illuminance of 700 lx, and noise level below 40 db. To minimize the influence of olfactory elements, an air purifier was placed within the room, and the door was periodically opened to allow fresh air circulation, ensuring any residual scents were minimized. Plants with strong natural fragrances were avoided or replaced with non-fragrant varieties. To control auditory distractions, soundproofing measures were implemented to maintain a noise level below 40 dB, and participants were provided with noise-canceling headphones when necessary. Ivory blackout curtains were installed to block visual elements inside the experimental space and create a sealed working space. Additionally, adjustable chairs that allowed comfortable learning were provided. For the tactile experiment, a white table measuring 180 cm × 90 cm × 70 cm (L × W × H) was used, along with a box (48 cm × 38 cm × 34 cm, L × W × H) with an opening and a concealed entrance to block visual stimuli.

## Experimental protocol

Before the experiment, all procedures were explained to participants to prevent unnecessary activities or communication. Participants entered the experimental space, wore the fNIRS device, and rested in a seated position for a 3-min baseline measurement to establish a stable state. Subsequently, participants engaged in tactile activities by touching 12 plants, classified into four tactile treatment groups (soft, smooth, stiff, and rough), for an average duration of 90 s each (Fig 1). After each activity, participants completed a self-report questionnaire based on the Semantic Differential Method (SDM). The experimental routine consisted of a 3-min rest, 90 s of activity, and a subsequent questionnaire, repeated for all 12 activities (Fig 2). The selection of plants involved a systematic process focused on specific sensory characteristics to ensure suitability for tactile stimulation. First, the definition of tactile sensation was verified using the Korean Standard Dictionary of National Language, establishing a baseline criterion

| Tactile Stimuli | Soft | Smooth | Stiff | Rough |
|---|---|---|---|---|
| 12 plants | *Stachys byzantina* | *Peperomia obtusifolia* | *Chamaeshparis thyoides Red Star* | *Rhapis excelsa* |
| | *Adiantum raddianum* | *Ficus benghalensis* | *Platycladus orientalis* | *Nephrolepis cordifolia 'Duffii'* |
| | *Asparagus plumosus var. nanus* | *Epipremnum aureum* | *Cupressus macrocarpa* | *Ardisia pusilla 'Variegata'* |

**Fig 1. 12plants used in the experiment are divided into four tactile stimulation groups.**

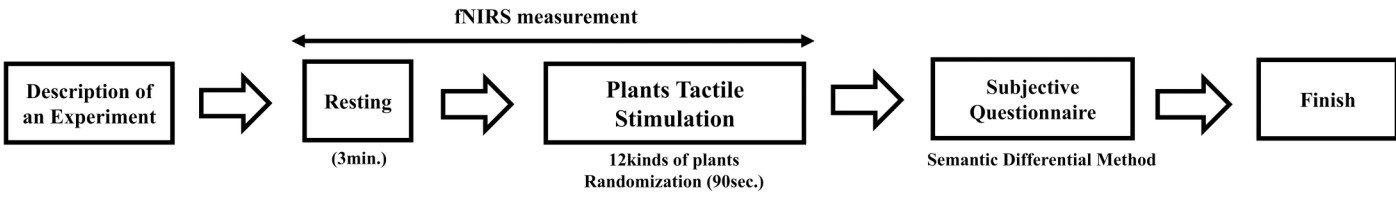

**Fig 2. Experiment protocol.**

for tactile classification. Plants were then classified by tactile sensation based on their botanical characteristics [31], as well as environmental factors influencing morphology and surface structure [32]. Further guidance was drawn from tactile classifications in wild plant data provided by the Korea National Arboretum [33]. To align the selected plants with participant preferences, foliage plant preference surveys [34] were referenced. This analysis helped identify plants that would be familiar and appealing to the participants. Additionally, risk factors associated with the plants were carefully reviewed [35], with any plant that could introduce tactile risks or non-tactile stimuli excluded to maintain a controlled tactile experience. Integrating these data sources enabled a robust classification of plants into four tactile categories: soft (3 types), smooth (3 types), stiff (3 types), and rough (3 types). This structured approach provided a diverse set of sensory experiences, essential for examining the psychophysiological and psychological responses to tactile plant interactions.

## Measurements

The fNIRS instrument used in this study was the NIRSIT Lite (OBELAB Inc., Seoul, Republic of Korea), covering the anterior PFC (CH 2-14), dorsolateral PFC (CH 1), and orbital part of the inferior frontal gyrus (CH15) within the Brodmann area (Fig 3). fNIRS is a noninvasive device that measures changes in cerebral blood flow using near-infrared light. It sends near-infrared light in the range of 700–1000 nm to the scalp, measures oxy-Hb and deoxy-Hb

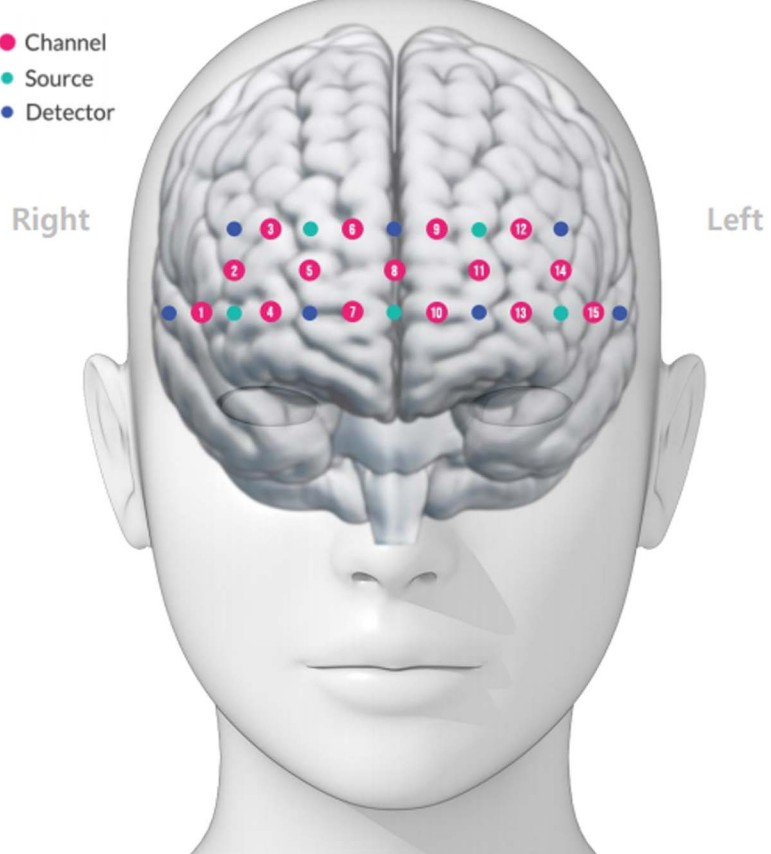

**Fig 3. fNIRS channel configuration on the prefrontal cortex.**

levels in the cerebral cortex in the transmitted area, and measures the concentration of oxygen in the blood [36]. Compared to real-time brain imaging devices such as functional magnetic resonance imaging or electroencephalogram, fNIRS is much simpler, less burdensome, and is less constrained by movement, environment, and space. Additionally, fNIRS provides both temporal and spatial resolution [37], making it advantageous for the real-time measurement of brain activation owing to its relatively high measurement speed [38]. Baseline data on oxy-Hb concentrations in the PFC were collected using fNIRS. Earlier studies indicated that oxy-Hb reflects cortical activation more effectively than deoxy-Hb when assessing hemoglobin levels in the PFC [39,40]. Changes in oxy-Hb concentrations were utilized to investigate differences in activity performance.

The SDM developed by Osgood (1952) evaluates subjective emotions using pairs of adjectives [41]. The SDM used in this study is a seven-point scale consisting of three items (uncomfortable–comfortable, tense–relaxed, and unfavorable–favorable).

## Data analysis

The fNIRS analysis used to identify the brain activation characteristics during the activity of touching twelve plants classified into four treatment groups (soft, smooth, stiff, and rough) was conducted as follows. For data preprocessing, raw data were subjected to low-pass filtering with a discrete cosine transform (DCT) applied at 0.1 and high-pass filtering applied with DCT at 0.0005. The signal-to-noise ratio (SNR) was calculated for data from 10 s to 15 s, and channels with an SNR of less than 30 dB were excluded from the analysis. Subsequently, the oxy-Hb concentration was calculated using the modified Beer–Lambert law (MBLL). The baseline for this calculation was obtained by averaging the first 90 s before the start of the task, and the task duration was set to an average of 90 s for the analysis. A one-way analysis of variance was conducted to test the differences between the twelve plants and the four types, followed by the post-hoc Duncan's multiple range test to rank the activities. Independent t-tests were performed to examine the differences between sexes among the four types. The statistical significance level for all activity analyses was set at $p < 0.05$. Statistical analyses were performed using IBM SPSS Statistics 26.0 (IBM Corp., Armonk, NY, USA).

## Results

### Demographic information

This study targeted 30 participants in their 20s, comprising an equal distribution of 50% males and 50% females. Characteristics of the participants are provided in Table 1.

**Table 1. Demographic information of the participants in this study.**

| Variance | Male (n = 15) | Female (n = 15) | Total (N = 30) |
|---|---|---|---|
| | Mean±SD | | |
| Age | 24.8 ± 2.33 | 23.93 ± 3.08 | 24.86 ± 2.68 |
| Height [1] (cm) | 173.57 ± 6.68 | 161.12 ± 4.97 | 167.34 ± 8.58 |
| Body Weight [2] (kg) | 70.52 ± 17.06 | 51.65 ± 8.49 | 61.08 ± 16.35 |
| Body mass index [3] (kg/m²) | 23.26 ± 4.17 | 19.91 ± 2.60 | 21.58 ± 3.82[4] |

[1] Height was measured using an anthropometer (Ok7979; Samhwa, Seoul, Republic of Korea) without shoes.

[2] Body weight was measured using a body fat analyzer (ioi 353; Jawon Medical, Republic of Korea).

[3] Body mass index was calculated using the formula: [weight (kg)]/ [height (m)²].

[4] Falls within the normal range proposed by the World Health Organization.

## Physiological responses

**fNIRS.** Among the four treatment groups (soft, smooth, stiff, and rough), the oxy-Hb concentration in the entire PFC was lowest during tactile stimulation with smooth plants and highest during stimulation with soft plants ($p < 0.05$). In the left PFC, the oxy-Hb concentration was the lowest during tactile stimulation with smooth plants and the highest during stimulation with soft plants ($p < 0.05$). In the right PFC, the oxy-Hb concentration was lowest during tactile stimulation with smooth and rough plants and highest during stimulation with soft plants ($p < 0.05$; Table 2).

When comparing the left and right PFC, the oxy-Hb concentration in the left PFC was lower than that in the right PFC during tactile stimulation with soft, smooth, and stiff plants, although the difference was not significant ($p > 0.05$). When touching rough plants, the oxy-Hb concentration in the right PFC was lower than that in the left PFC, although the difference was not statistically significant ($p > 0.05$; Table 3).

Among the four treatment groups (soft, smooth, stiff, and rough), men showed a significantly lower oxy-Hb concentration in the PFC area during tactile stimulation with smooth, stiff, and rough plants, and the highest concentration during tactile stimulation with the soft plant ($p < 0.001$). By contrast, no significant difference was observed in oxy-Hb concentration in the PFC of women ($p > 0.05$). When touching soft plants, the oxy-Hb concentration in the overall PFC area of women was significantly lower than that in men ($p < 0.001$). When

**Table 2. Comparison of prefrontal cortex oxyhemoglobin concentration by tactile stimulation (Duncan's post-hoc test).**

| Tactile Stimuli | Entire PFC [1] | Left PFC [2] | Right PFC [3] |
|---|---|---|---|
| | Mean±SD (mM) | | |
| Soft plants | $(1.25 \pm 3.06) \times 10^{-4}$ b | $(2.59 \pm 3.56) \times 10^{-4}$ b | $(4.3 \pm 4.6) \times 10^{-4}$ b |
| Smooth plants | $(-0.88 \pm 3.23) \times 10^{-4}$ a | $(-0.97 \pm 2.21) \times 10^{-4}$ a | $(0.61 \pm 3.57) \times 10^{-4}$ a |
| Stiff plants | $(0.3 \pm 3.76) \times 10^{-4}$ ab | $(4.07 \pm 0.88) \times 10^{-4}$ ab | $(1.61 \pm 6.21) \times 10^{-4}$ ab |
| Rough plants | $(0.33 \pm 3.76) \times 10^{-4}$ ab | $(0.44 \pm 4.07) \times 10^{-4}$ ab | $(0.06 \pm 3.71) \times 10^{-4}$ a |
| F | 2.873 | 3.264 | 3.485 |
| *p*-value | 0.038* | 0.026* | 0.02* |

NS, *, ** non-significant or significant at $p < 0.05$ and 0.01, respectively, using one-way analysis of variance.

The statistical method used Duncan's post-hoc analysis (a > b).

Lowercase letters indicate the group to which the activity belonged when performing an analysis using the Duncan's test.

[1] Entire PFC refers to CH1–15 on NIRSIT LITE.

[2] Left PFC refers to CH 8–15 on NIRSIT LITE.

[3] Right PFC refers to CH 1–7 on NIRSIT LITE.

**Table 3. Comparison of oxyhemoglobin concentrations between the left and right prefrontal cortex.**

| Tactile Stimuli | Left PFC [1] | Right PFC [2] | t | *p*-value |
|---|---|---|---|---|
| | Mean±SD (mM) | | | |
| Soft plants | $(2.59 \pm 3.56) \times 10^{-4}$ | $(4.3 \pm 4.62) \times 10^{-4}$ | -1.265 | 0.22 NS |
| Smooth plants | $(-0.97 \pm 2.21) \times 10^{-4}$ | $(0.61 \pm 3.57) \times 10^{-4}$ | -1.903 | 0.072 NS |
| Stiff plants | $(0.46 \pm 4.63) \times 10^{-4}$ | $(1.61 \pm 6.12) \times 10^{-4}$ | -0.591 | 0.561 NS |
| Rough plants | $(0.44 \pm 4) \times 10^{-4}$ | $(0.06 \pm 3.71) \times 10^{-4}$ | 0.255 | 0.801 NS |

NS, non-significant at $p > 0.05$, using paired t-tests.

[1] Left PFC refers to CH 8-15 on NIRSIT LITE.

[2] Right PFC refers to CH 1-7 on NIRSIT LITE.

touching smooth, stiff, and rough plants, the oxy-Hb concentration in women was lower than that in men; however, the difference was not significant ($p > 0.05$; Table 4).

## Psychological responses

**SDM.** Among the four treatment groups (soft, smooth, stiff, and rough), participants indicated that tactile stimulation with soft and smooth plants induced the highest levels of relaxation ($p < 0.001$). Additionally, the participants reported that tactile stimulation with soft and smooth plants was associated with the highest level of comfort ($p < 0.001$). Participants expressed that tactile stimulation with soft and smooth plants was the most favorable among all four treatment groups ($p < 0.001$; Fig 4).

Among the four treatment groups (soft, smooth, stiff, and rough), men responded that tactile stimulation with soft and smooth plants induced the highest level of relaxation ($p < 0.001$), whereas women responded that tactile stimulation with smooth plants induced the highest level of relaxation ($p < 0.001$). When comparing relaxation scores based on sex, a significant difference was observed among the soft plants ($p < 0.01$), although no significant differences were observed among the other three treatment groups (smooth, stiff, and rough; $p > 0.05$). Men reported that tactile stimulation with soft plants induced the highest level of comfort ($p < 0.001$), whereas women reported that tactile stimulation with smooth plants induced the highest level of comfort ($p < 0.001$). No significant difference was observed in the comfort scores based on sex ($p > 0.05$). Men reported that tactile stimulation with soft, smooth, and rough plants was the most favorable ($p < 0.001$), whereas women reported that tactile stimulation with smooth plants was the most favorable ($p < 0.001$). When comparing response scores for the favorable items based on sex, a significant difference was observed between soft ($p < 0.01$) and stiff ($p < 0.05$) plants, whereas no significant difference was observed between the other two treatment groups (smooth and rough) ($p > 0.05$; Fig 5A and 5B).

## Discussion

In this study, we examined the physiological and psychological responses to tactile stimulation with different plant textures, specifically focusing on changes in oxyhemoglobin (oxy-Hb) concentrations in the prefrontal cortex (PFC) and participants' subjective assessments. Our findings reveal distinct psychophysiological effects associated with each tactile group (soft,

**Table 4. Comparison of the oxyhemoglobin concentrations in the prefrontal cortex by sex.**

| Tactile Stimuli | Entire PFC [1] Men (n = 15) | Entire PFC [1] women (n = 15) | t | *p*-value |
|---|---|---|---|---|
| | Mean±SD (mM) | | | |
| Soft plants | $(3.36 \pm 4.07) \times 10^{-4}$ b | $(-0.81 \pm 3.98) \times 10^{-4}$ | 4.916 | 0.000*** |
| Smooth plants | $(-0.22 \pm 2.97) \times 10^{-4}$ a | $(-1.48 \pm 5.16) \times 10^{-4}$ | 1.418 | 0.161 NS |
| Stiff plants | $(1.09 \pm 5.27) \times 10^{-4}$ a | $(-0.45 \pm 3.66) \times 10^{-4}$ | 1.614 | 0.11 NS |
| Rough plants | $(0.36 \pm 3.89) \times 10^{-4}$ a | $(0.33 \pm 4.49) \times 10^{-4}$ | 0.036 | 0.971 NS |
| F | 6.496 | 1.291 | | |
| *p*-value | 0.000*** | 0.279 NS | | |

NS, *** non-significant or significant at P < 0.001, respectively, as determined using one-way analysis of variance.

Duncan's post-hoc test was used for statistical analyses (a > b). Lowercase letters indicate the group to which the activities belong when performing analysis using Duncan's test.

NS, *** non-significant, or significant at $p < 0.001$, respectively, using independent t-tests.

[x] Entire PFC refers to CH1–15 on NIRSIT LITE.

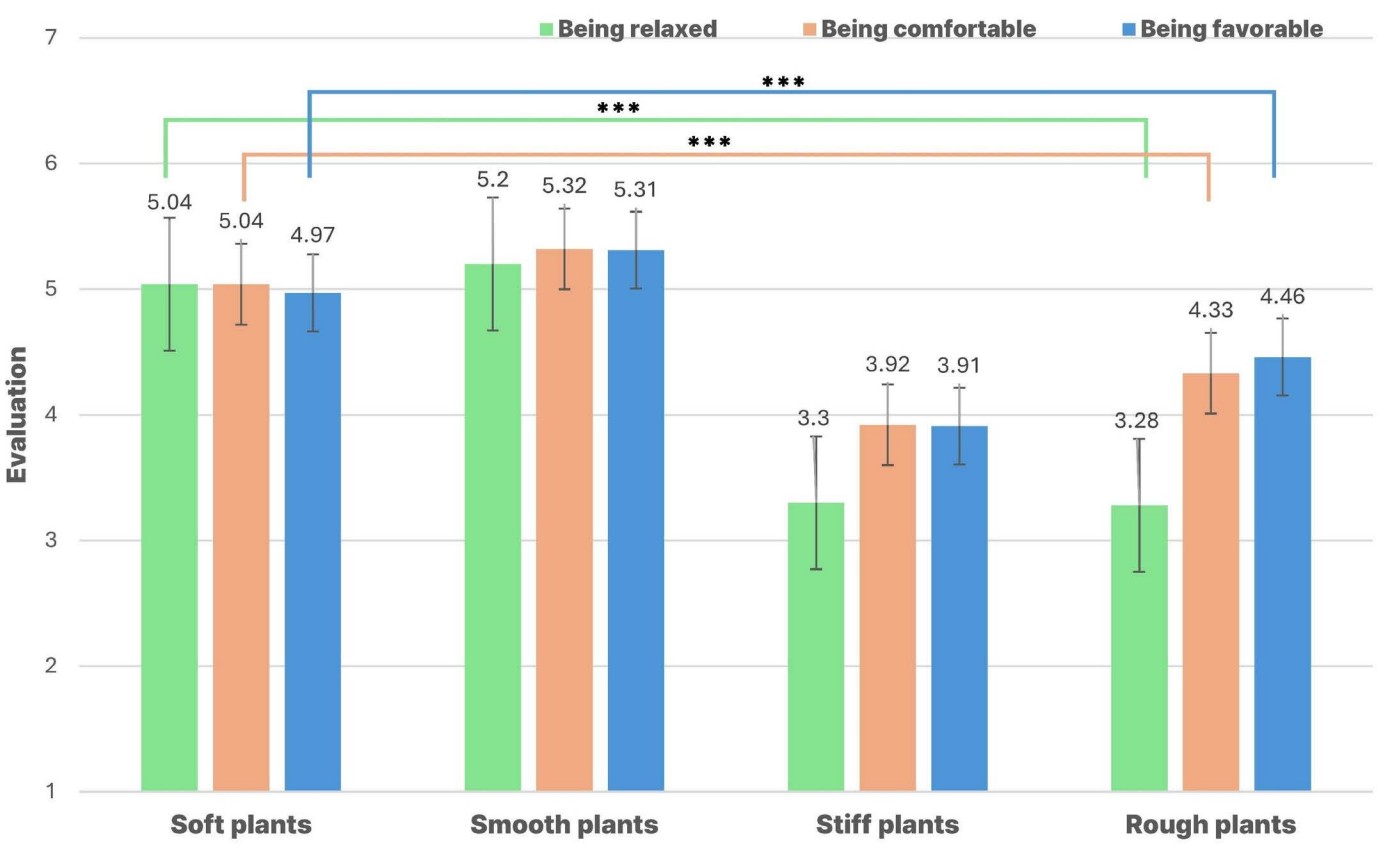

**Fig 4. Comparison of SDM during tactile stimulation.**

smooth, stiff, and rough) demonstrating that plant texture significantly influences well-being and mental responses.

Our physiological easurements indicate that smooth plant textures are associated with the lowest oxy-Hb concentration in the PFC, while soft plant textures elicit the highest concentration ($p < 0.05$). These results align with prior research suggesting that smooth textures generally promote a calming effect, possibly by reducing neural activation in the PFC [42]. Studies on tactile perception reveal that smooth surfaces often require less complex cognitive engagement for processing, resulting in fewer demands on attentional resources and facilitating a state of relaxation [43]. The observed decrease in oxy-Hb concentration aligns with findings from sensory psychology that demonstrate how smooth, familiar textures can activate relaxation-related neural pathways, reducing the load on the PFC [44,45]. In contrast, soft textures with fine, hair-like features appear to increase neural activation in the PFC, potentially due to the sensory processing demands involved in interpreting these more nuanced tactile characteristics [46]. This heightened response is consistent with findings in haptics research, which indicate that hair-like or fuzzy textures stimulate areas of the brain linked to emotional processing and attention, as these surfaces are less predictable and require more complex sensory discrimination [47,48]. These findings indicate that the sensory demands of different textures can either amplify or reduce PFC activity, highlighting the nuanced role of texture in physiological responses.

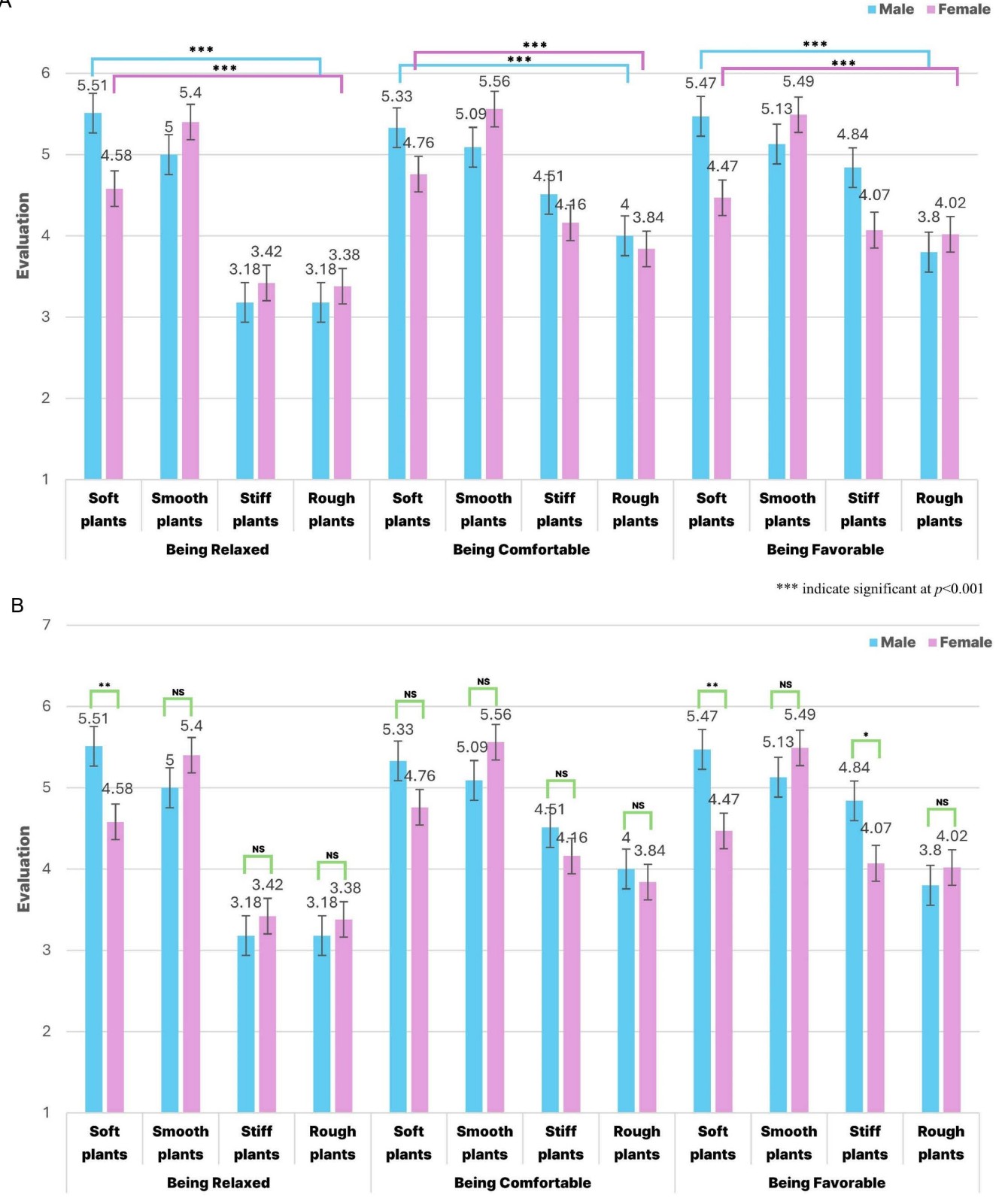

**Fig 5. (A) Variation in SDM responses based on sex (B) Comparison of responses between sex.**

Sex-based differences in physiological responses also emerged. Male participants exhibited significantly lower oxy-Hb concentrations when interacting with smooth, stiff, and rough plants compared to soft plants (p < 0.001). This response suggests that males may process these textures with reduced neural activation in the PFC, potentially due to differences in cognitive or sensory processing mechanisms associated with familiarity or past tactile experiences [49]. Studies suggest that males and females may engage in sensory experiences differently, with males often exhibiting lower neural activation for familiar tactile stimuli, which could stem from varied neural circuitry and sensory preferences developed over time [50]. This may be influenced by socio-cultural factors, as tactile interactions with different textures are often learned and reinforced through experiences shaped by gender roles and societal expectations [51]. In contrast, females showed a notable reduction in oxy-Hb concentration when interacting with soft plants compared to males (p < 0.001). This response could reflect a heightened emotional sensitivity to soft textures, which tends to stabilize PFC activity due to an affective response to the texture's comforting properties. Research indicates that females often exhibit a stronger physiological and emotional response to soft textures, potentially linked to greater affective engagement during tactile experiences [52]. Prior studies in sensory science support this observation, as individuals who favor softer stimuli often report higher emotional intensity and relaxation, suggesting that softer textures provide a form of sensory comfort that can lead to reduced neural activation [53,54]. These sex-based differences underscore the importance of considering individual factors, such as gender, previous tactile experiences, and personal preferences, in understanding responses to tactile plant interactions [55].

Psychological assessments using the SDM provided additional insights that reinforce our physiological findings. Participants reported experiencing the highest levels of relaxation, comfort, and favorability when interacting with soft and smooth plants. Men reported that soft plants were the most comfortable (p < 0.001), whereas women found smooth plants to be the most comforting. These findings align with existing literature on the calming and favorable psychological effects of smooth and soft textures. In sensory science, soft textures are frequently associated with comfort and warmth, while smooth textures often elicit perceptions of harmony and tranquility [56,57]. These psychological associations appear to correlate with lower neural demands, as evidenced by the oxy-Hb concentration findings, and suggest a physiological basis for the subjective feelings of comfort and relaxation reported by participants. Interestingly, the link between lower oxy-Hb levels and favorable SDM scores in both smooth and soft textures highlights a measurable neural basis for these subjective responses, suggesting that these textures could play a role in regulating emotional states through minimal neural activation [58,59]. In contrast, rough textures elicited lower favorability ratings. This response aligns with findings in sensory and psychological research that associate rough or abrasive textures with heightened sensory awareness and discomfort, potentially due to the irregular nature of these surfaces, which may demand greater attentional resources and lead to neutral or less positive psychological responses [60,61]. Rough textures often activate neural pathways related to vigilance and awareness, possibly leading to less relaxation and higher neural activation in regions associated with alertness and cognitive processing [62]. Such insights reinforce the notion that tactile qualities can evoke distinct psychological responses, adding further evidence to the potential therapeutic value of specific plant textures in applications aimed at mental relaxation and emotional stability [63]. The observed psycho-physiological effects of plant texture provide a foundation for integrating these findings into sensory-focused therapeutic applications [64]. Tactile plant interactions in therapeutic settings could be tailored based on individual preferences and sensitivities [65], providing a customized approach that aligns with both physiological and psychological needs. Given that tactile experiences with specific textures can influence both physiological markers of relaxation and

subjective comfort, there is a promising potential for developing sensory-based interventions that maximize therapeutic outcomes for diverse populations [66].

While this study centered on the physiological and psychological effects of tactile stimulation, future research could expand our understanding of plant-based interactions through a multisensory approach. Combining visual and tactile stimuli may reveal potential synergies between sensory modalities, offering a more comprehensive view of how plant interactions affect mental well-being. Investigating visual-tactile interactions could shed light on neural responses, particularly given the interest in multisensory therapeutic applications for mental health. Additionally, while this study focused on the PFC's role in emotional regulation, future research might explore other brain regions, such as the amygdala or posterior sensory areas, to assess how texture impacts broader emotional processing. Mapping responses across multiple brain regions could deepen insights into the restorative effects of plant-based therapies, supporting sensory-focused horticultural interventions.

## Conclusion

This study highlights the complex psychophysiological effects of tactile engagement with plants, demonstrating that variations in foliage texture can produce distinct physiological and psychological responses that vary by plant type and sex. Our results show that tactile interaction with smooth and soft plants induces different levels of neural activity and subjective comfort, underscoring the role of texture in influencing emotional stability and mental relaxation. Specifically, the reduction in oxy-Hb concentrations in certain areas of the PFC, combined with participants' subjective feelings of relaxation and comfort, points to the potential for tactile plant interactions to promote psychological well-being and stability. These findings also underscore the therapeutic value of plant textures, as certain textures appear to foster psychological relaxation and reduced neural activity, laying a foundation for sensory-based interventions in horticultural therapy. Integrating tactile plant interactions in therapeutic settings could help design interventions tailored to individual preferences and physiological responses. To further support these findings, future studies should incorporate multisensory approaches and examine responses across diverse brain regions. Such research is expected to contribute to evidence-based mechanisms for horticultural therapies, enhancing the applicability of plant-based interventions in promoting mental health.

## Supporting information

**S1 File. Minimal data set.**
(XLSX)

**S2 File.**
(DOCX)

## Acknowledgments

This study was supported by the KU Research Professor Program at Konkuk University. We express our sincere gratitude to Mi-Sook Jeong and Seo-Yeon Park for their invaluable contribution as research experimental assistants. Their support and dedication significantly enhanced the quality of our work.

## Author contributions

**Conceptualization:** Seo-Hyun Kim, Sin-Ae Park.

**Data curation:** Seo-Hyun Kim, Sin-Ae Park.

**Formal analysis:** Seo-Hyun Kim, Sin-Ae Park.

**Funding acquisition:** Sin-Ae Park.

**Investigation:** Seo-Hyun Kim, Sin-Ae Park.

**Methodology:** Seo-Hyun Kim, Sin-Ae Park.

**Project administration:** Seo-Hyun Kim, Sin-Ae Park.

**Resources:** Seo-Hyun Kim, Sin-Ae Park.

**Supervision:** Sin-Ae Park.

**Validation:** Seo-Hyun Kim, Sin-Ae Park.

**Visualization:** Seo-Hyun Kim, Sin-Ae Park.

**Writing – original draft:** Seo-Hyun Kim, Sin-Ae Park.

**Writing – review & editing:** Seo-Hyun Kim, Sin-Ae Park.

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
