## [Decision Letter · Decision Letter 0]

16 Oct 2024

Dear Dr. Park,

Thank you for submitting your manuscript to PLOS ONE. After careful consideration, we feel that it has merit but does not fully meet PLOS ONE’s publication criteria as it currently stands. Therefore, we invite you to submit a revised version of the manuscript that addresses the points raised during the review process.

Upon reviewing the manuscript titled *Psychophysiological and psychological responses of touching plant behavior by tactile stimulation according to the foliage type* , I found the following points regarding language and content clarity:

<h3>Language and Readability:</h3>

**Grammar and Syntax** : The manuscript's language is clear, with generally good sentence structure and correct grammar. However, there are some areas where the sentence construction could be made smoother. For example, some sentences are lengthy and would benefit from being broken down into shorter, clearer segments.**Word Choice and Repetition** : The text sometimes repeats phrases like "tactile stimulation" excessively. Varying terminology could enhance the readability.**Scientific Terminology** : The technical terms are used appropriately, but there are instances where simpler explanations could make the manuscript more accessible to a broader audience.

<h3>Suggestions for Improvement:</h3>

**Clarity in Method Descriptions** : The section on the experimental protocol is detailed but could benefit from clearer transitions between the steps. Including a flow diagram might help clarify the procedure for readers.**Results Section** : The explanation of the statistical methods and results is clear but can be overwhelming due to the dense presentation of numbers. Using tables and figures more strategically could improve readability.**Discussion Section** : The discussion could be expanded to include more comparisons with existing literature, especially about the underlying mechanisms behind the physiological responses to different plant types.

<h3>Native Language Review:</h3>

Although the manuscript is well-written, a review by a native English speaker specializing in academic writing could further enhance the flow and reduce any lingering awkwardness in phrasing.

In summary, the manuscript is understandable, but refining sentence structure, varying terminology, and polishing the flow could improve the overall presentation. A native English speaker might add value by enhancing readability and clarity.

We look forward to receiving your revised manuscript.

Kind regards,

Zahra Lorigooini

Academic Editor

PLOS ONE

Journal Requirements:

“This work was carried out with the support of the “Cooperative Research Program for Agriculture Science and Technology Development (Project No.:RS-2021-RD009877)”, Rural Development Administration, Republic of Korea.”

“This work was carried out with the support of “Cooperative Research Program for Agriculture Science and Technology Development (Project No.:RS-2021-RD009877)” Rural Development Administration, Republic of Korea.

We express our sincere gratitude to Mi-Sook Jeong and Seo-Yeon Park for their invaluable contribution as a research experimental assistant. Their support and dedication significantly enhanced the quality of our work.”

“This work was carried out with the support of the “Cooperative Research Program for Agriculture Science and Technology Development (Project No.:RS-2021-RD009877)”, Rural Development Administration, Republic of Korea.”

Reviewers' comments:

Reviewer's Responses to Questions

**Comments to the Author**

1. Is the manuscript technically sound, and do the data support the conclusions?

Reviewer #1: Partly

Reviewer #2: Yes

2. Has the statistical analysis been performed appropriately and rigorously?

Reviewer #1: Yes

Reviewer #2: Yes

3. Have the authors made all data underlying the findings in their manuscript fully available?

Reviewer #1: Yes

Reviewer #2: Yes

4. Is the manuscript presented in an intelligible fashion and written in standard English?

Reviewer #1: Yes

Reviewer #2: Yes

Reviewer #1: Authors submitted a study of tactile response on a human wellbeing by using different plant species divided to different subgroups that shall evoke different behavioural response. Despite the idea is interesting, the present form of the text lack scientific integration of the meaningful references linked to their results from the study.

Introduction: The manuscript makes broad claims on relation of urbanization to stress claiming "high levels of urbanization and artificial environments exacerbate human stress levels". The text shall focus on measurements and results rather than to general foresight conclusions.

The manuscript writes on theories like the attention restoration theory and psychological-evolutionary theory but does not integrate enough these theories into the rationale and methods used in the study. A discussion of how these theories specifically relate to tactile stimulation would strengthen the scientific background.

The manuscript’s introduction and literature review sections lack seamless integration, where the flow of ideas does not always logically connect one to the next.

It seems that all references are not precisely cited according to the findings of the cited studies. Some might be misleading (6, 8, …)? Also, the manuscript refers to the references in a way that suggests a closer relationship to the current research than is warranted or is used to support a broad statement without detailing how these benefits directly tie to the present results. Also, very general statements are somewhere connected to the neuroscientific or genetic background or mental health benefits, despite not supported by strong evidence from the results.

Some references are used to support multiple claims without clear indication of how each reference specifically supports each claim, leading to potential confusion and perceived "mix-up" of references.

In Result section at least the graphic/picture presentation in the brain areas will improve understanding the main findings.

The research predominantly focuses on tactile sensation and does not evaluate multisensory approaches comprehensively, e.g., visual, auditory to better understand the impact of plant interactions on human well-being. Despite that, rewritten manuscript may eliminate this lacking results.

Reviewer #2: Psychophysiological and psychological responses of touching plant behavior by tactile

stimulation according to the foliage type

This study measured and compared the psychophysiological and psychological responses to tactile stimuli through plant contact based on the foliage type. The experimental design is sound, and the methodology is clearly described, although a few areas could benefit from additional explanation or revision. Overall, the study's findings report evaluable results in Responses to touching plant behavior by tactile stimulation.

Here are some suggestions for this study:

1. In the abstract, a section should be dedicated to Background.

2. There are some writing errors throughout the manuscript. Please revise all text. For example, in line 12 we aimed to measure...

3. In the Experimental protocol section, Add more information about plants used in this study. e.g. morphological characteristics of the leaves, whether the plants are fragrant or not, compounds, and whether they may have compounds absorbed by the skin through touch...

4. In Fig. 1, the figures of plants are not clear. Provide better quality images.

5. In the Experimental environment section, what conditions were controlled regarding the effect of the smell element on the test conditions?

6. Duncan's test should be mentioned in the subtitle of the table 3., not using one-way analysis of variance

7. Why is there no control group in this study?

**Do you want your identity to be public for this peer review?** For information about this choice, including consent withdrawal, please see our Privacy Policy

Reviewer #1: No

Reviewer #2: No

---

## [Author Response · Author response to Decision Letter 1]

11 Nov 2024

Dear Editors and Reviewers,

We would like to extend our sincere thanks for the constructive feedback provided on our manuscript. We have carefully reviewed each comment and revised address the concerns raised. Below, we provide responses to each point raised by the reviewers, as well as details on how we have revised the manuscript.

Please note that all changes are highlighted using the tracked changes feature in the revised Word file, allowing reviewers to view our modifications directly.

Reviewer 1

1. Comment: "Authors submitted a study of tactile response on human well-being by using different plant species divided into different subgroups that shall evoke different behavioral responses. Despite the idea is interesting, the present form of the text lacks scientific integration of the meaningful references linked to their results from the study."

Response: Thank you for this feedback. We agree that incorporating relevant references to support our results will strengthen the scientific integration of our study. We have added more references in the Introdcution and Discussion sections to link our findings with the existing literature and have ensured that each reference accurately supports the findings discussed.

2. Comment: "Introduction: The manuscript makes broad claims on the relation of urbanization to stress claiming 'high levels of urbanization and artificial environments exacerbate human stress levels.' The text shall focus on measurements and results rather than general foresight conclusions."

Response: We appreciate your input on maintaining focus on the specific study context. We have revised the introduction to limit broad statements about urbanization and stress and instead emphasize studies on stress-related measurements and direct relevance to tactile interactions with plants.

3. Comment: "The manuscript writes on theories like the attention restoration theory and psychological-evolutionary theory but does not integrate enough these theories into the rationale and methods used in the study. A discussion of how these theories specifically relate to tactile stimulation would strengthen the scientific background."

Response: We have removed the references to the attention restoration and psychological-evolutionary theories, as they do not directly connect to the results of our study. Instead, we have added new literature that more effectively strengthens the introduction, providing a clearer foundation for the study’s rationale and methods by focusing specifically on tactile stimulation and its established role in supporting mental and physiological health.

4. Comment: "The manuscript’s introduction and literature review sections lack seamless integration, where the flow of ideas does not always logically connect one to the next."

Response: We appreciate this observation and have reorganized portions of the introduction and literature review to create a more logical flow. We have made transitions between ideas more explicitly, ensuring each section builds on the previous one, leading seamlessly into the study’s rationale.

5. Comment: "It seems that all references are not precisely cited according to the findings of the cited studies. Some might be misleading (6, 8, …)? Also, the manuscript refers to the references in a way that suggests a closer relationship to the current research than is warranted."

Response: We understand the importance of accurate citation and clear linkage of references to specific claims. We reviewed each reference to verify its relevance and have clarified how each cited work supports the points in our study. We also replaced or revised references where necessary to ensure accuracy and avoid any misleading implications.

6. Comment: "Also, very general statements are somewhere connected to the neuroscientific or genetic background or mental health benefits, despite not supported by strong evidence from the results."

Response: Thank you for highlighting this. We have removed or modified overly general statements regarding neuroscience or mental health benefits that were not directly supported by our study's results and instead focused on findings that are directly relevant to the data collected.

7. Comment: "Some references are used to support multiple claims without clear indication of how each reference specifically supports each claim, leading to potential confusion and perceived 'mix-up' of references."

Response: We reviewed the manuscript and clarified how each reference supports specific claims. Where a reference is used to support multiple claims, we have clarified each application separately to avoid any potential confusion and ensure each use of the reference is clear.

8. Comment: "In the Result section at least the graphic/picture presentation in the brain areas will improve understanding of the main findings."

Response: We appreciate this suggestion to enhance clarity. In the Results section, we added diagrams showing brain areas relevant to the prefrontal cortex and indicated the focal regions for oxyhemoglobin concentration measurement. This visual aid will help readers understand the physiological context of the findings.

9. Comment: "The research predominantly focuses on tactile sensation and does not evaluate multisensory approaches comprehensively, e.g., visual, auditory to better understand the impact of plant interactions on human well-being."

Response: We acknowledge that a multisensory approach could indeed offer a broader perspective on plant interactions. In the Discussion section, we have noted the limitations of focusing solely on tactile sensation and suggested that future studies could explore multisensory approaches to provide a more comprehensive assessment. Given the current lack of research specifically on plant tactile stimuli, we believe it is essential to focus on this aspect first before extending to multisensory measurements. The manuscript has been revised to emphasize this focused approach while acknowledging the limitation.

Reviewer 2

1. Comment: "In the abstract, a section should be dedicated to Background."

Response: We appreciate this suggestion and agree that adding a background section to the abstract would enhance context for readers. We have revised the abstract to include a brief background that outlines the study’s purpose and relevance to urban stress relief and nature-based therapies through tactile plant interactions.

2. Comment: "There are some writing errors throughout the manuscript. Please revise all text. For example, in line 12 we aimed to measure..."

Response: We have carefully reviewed the entire manuscript for grammar and syntax issues, paying particular attention to areas with awkward phrasing or errors, including the example provided. Revisions have been made to ensure clarity, readability, and grammatical correctness throughout the document.

3. Comment: "In the Experimental Protocol section, add more information about plants used in this study. E.g., morphological characteristics of the leaves, whether the plants are fragrant or not, compounds, and whether they may have compounds absorbed by the skin through touch."

Response: Thank you for this insightful suggestion. We have added detailed descriptions of the plant species used in the study, including leaf morphology, fragrance characteristics, and potential compounds relevant to skin absorption. This additional information aims to provide a more comprehensive understanding of each plant’s tactile properties and any factors that might affect the results.

4. Comment: "In Fig. 1, the figures of plants are not clear. Provide better quality images."

Response: We apologize for the quality of the images in Figure 1. We have replaced these with higher-resolution images to ensure clarity and improved visual quality, which should help readers identify the tactile characteristics of each plant more easily.

5. Comment: "In the Experimental Environment section, what conditions were controlled regarding the effect of the smell element on the test conditions?"

Response: To control olfactory stimuli, we implemented several measures to minimize the potential influence of scent during the experiment. An air purifier was used in the room, and the door was periodically opened to allow for air circulation. Additionally, we avoided using plants with strong natural fragrances, selecting non-fragrant varieties wherever possible. We have now detailed these conditions in the manuscript to clarify the methods used to control smell and focus exclusively on tactile stimuli.

6. Comment: "Duncan's test should be mentioned in the subtitle of Table 3, not using one-way analysis of variance."

Response: We have revised the subtitle of Table 3 to specify that Duncan’s post-hoc test was used, addressing this suggestion. This clarification aligns with the statistical analysis performed and improves the accuracy of the table description.

7. Comment: "Why is there no control group in this study?"

Response: The primary objective of this study was to assess the psychophysiological and psychological responses to different tactile experiences with plant foliage (soft, smooth, stiff, and rough). Given this objective, each plant type itself served as a distinct condition in a within-subjects design, where each participant experienced all conditions for direct comparison. Therefore, a separate control group was deemed unnecessary.

Instead of comparing the presence or absence of tactile plant interaction (which would typically require a control group), we focused on comparing the different tactile textures directly. To ensure validity in the comparisons, we obtained baseline measurements of oxyhemoglobin (oxy-Hb) concentration in the prefrontal cortex (PFC) and psychological states before each tactile experience. These baseline measurements allowed each participant to serve as their own control across conditions, ensuring that the responses could be accurately attributed to the tactile differences among the plant types.

This approach allowed us to focus on relative differences between the plant types and assess how specific tactile textures influence psychophysiological and psychological responses without introducing an external control group. We have clarified this rationale in the manuscript to provide additional context.

We hope these revisions and clarifications meet with your approval, and we are grateful for your time and effort in helping us enhance the quality of this study. Thank you again for your insightful and constructive feedback.

Sincerely,

---

## [Decision Letter · Decision Letter 1]

16 Dec 2024

Psychophysiological and psychological responses of touching plant behavior by tactile stimulation according to the foliage type

PONE-D-24-07959R1

Dear Dr. Park,

We’re pleased to inform you that your manuscript has been judged scientifically suitable for publication and will be formally accepted for publication once it meets all outstanding technical requirements.

Kind regards,

Zahra Lorigooini

Academic Editor

PLOS ONE

Additional Editor Comments (optional):

Reviewers' comments:

Reviewer's Responses to Questions

**Comments to the Author**

Reviewer #2: (No Response)

2. Is the manuscript technically sound, and do the data support the conclusions?

Reviewer #2: Yes

3. Has the statistical analysis been performed appropriately and rigorously?

Reviewer #2: Yes

4. Have the authors made all data underlying the findings in their manuscript fully available?

Reviewer #2: Yes

5. Is the manuscript presented in an intelligible fashion and written in standard English?

Reviewer #2: (No Response)

Reviewer #2: (No Response)

**Do you want your identity to be public for this peer review?** For information about this choice, including consent withdrawal, please see our Privacy Policy

Reviewer #2: No

---

## [Editor Report · Acceptance letter]

PONE-D-24-07959R1

PLOS ONE

Dear Dr. Park,

I'm pleased to inform you that your manuscript has been deemed suitable for publication in PLOS ONE. Congratulations! Your manuscript is now being handed over to our production team.

Kind regards,

on behalf of

Dr. Zahra Lorigooini

Academic Editor

PLOS ONE